# Overweight and Obese Children Aged 6–17 Years in China Had Lower Level of Hydration Status: A Cross-Sectional Study

**DOI:** 10.3390/nu17020364

**Published:** 2025-01-20

**Authors:** Jianfen Zhang, Wei Cao, Juan Xu, Hongliang Wang, Ruihe Luo, Qian Gan, Titi Yang, Hui Pan, Zhenyu Yang, Wenhua Zhao, Qian Zhang

**Affiliations:** 1National Institute for Nutrition and Health, Chinese Center for Disease Control and Prevention, 27 Nanwei Road, Xicheng District, Beijing 100050, China; zhangjf@ninh.cdc.cn (J.Z.); caowei@ninh.chinacdc.cn (W.C.); xujuan@ninh.chinacdc.cn (J.X.); wanghl@ninh.chinacdc.cn (H.W.); luorh@ninh.chinacdc.cn (R.L.); ganqian@ninh.chinacdc.cn (Q.G.); yangtt@ninh.chinacdc.cn (T.Y.); panhui@ninh.chinacdc.cn (H.P.); yangzy@ninh.chinacdc.cn (Z.Y.); zhaowh@chinacdc.cn (W.Z.); 2Key Laboratory of Public Nutrition and Health, National Health Commission of the People’s Republic of China, 27 Nanwei Road, Xicheng District, Beijing 100050, China

**Keywords:** total body water, extracellular water, intracellular water, hydration, health, children

## Abstract

Purpose: The aims of this study were to explore the differences in total body water and hydration status among Chinese children aged 6–17 years. Methods: A cross-sectional study was implemented among children aged 6–17 years in China. The total body water (TBW), intracellular water (ICW), and extracellular water (ECW) were determined by bioelectrical impedance analysis (BIA). The participants were divided according to age—age 6–8 years, age 9–11 years, age 12–14 years, age 15–17 years—and body mass index (BMI) of China—underweight, normal weight, overweight, and obese groups. The differences of variables of groups were compared using analysis of variance, Student’s t-test, and Kruskal–Wallis test. Significance levels were set at 0.05 (*p* < 0.05). Results: A total of 59,643 participants (30,103 males and 29,540 females) completed the study. As children became older, the TBW, ICW, ECW, ICW/TBW, and TBW/FFM (TBW to fat free mass ratio) increased simultaneously (all *p* < 0.05); concurrently, the ECW/TBW decreased with age (all *p* < 0.05). Boys had higher TBW, ICW, ECW, ICW/TBW, TBW/BW, and TBW/FFM than those of girls at each age (all *p* < 0.05). For all BMI groups, increases in TBW, ICW, ECW were observed from the underweight group to the obese group, both in boys and girls (all *p* < 0.001). For the increase in BMI in all age groups, the values of TBW made a significantly lower percentage compared to BW. The higher BMI groups showed higher levels of TBW/FFM, both in girls and boys (all *p* < 0.001). Conclusions: The body water contents of children aged 6–17 years varied according to their age, sex, and BMI. Overweight and obese individuals may have inferior hydration status compared to those with normal weight.

## 1. Background

Water is not only an essential nutrient but also a key molecule essential for keeping cellular homoeostasis, which accounts for 50–75% of the body weight (BW). The body’s water content consists of intracellular water (ICW) and extracellular water (ECW) [1]. ICW compromises 2/3 of the total body water (TBW), which is the principal element influencing the cell volume. Meanwhile, the ECW consists of plasma, interstitial fluid, and other transcellular fluids [2]. Maintaining body dynamic water balance between ICW and ECW is a matter of public health concern, especially in infants and young children [3]. It has been proved that body water compartments are impacted by water intake. Given that children have higher proportions of total water content in the body and are less sensitive to thirst sensation, children are more prone to have inadequate water intake than adults. Accordingly, children with inadequate water intake have more risk to be dehydrated than those meeting the recommendation of total water intake, which could lead to adverse effects on health [4,5]. Therefore, it is of great significance to have adequate water intake and maintain the dynamic water balance of the body in children.

More generally, the TBW and its compartments are associated with health. Currently, TBW is used to determine the hydration status [6,7,8]. Hydration statuses are divided into dehydration, optimal hydration, and middle hydration status. Dehydration, as a deficit in TBW, is associated with adverse health outcomes among children, including physical performances, cognition, intestinal mucus homeostasis, sleep duration, faster aging, and even has been linked to obesity and malnutrition [9,10,11,12]. Additionally, the proportion of ECW to TBW utilized is a signifier of the human body’s volume overload. Meanwhile, studies have revealed that the water content of the body is not only correlated with hydration status but also with health [13,14,15,16,17,18]. In the case of healthy people, in comparison with youth, the ICW compartment proved to be a more reliable predictor for muscle performance of both adults and older adults [17]. Regarding people with diseases, measuring body water content provided clinically valuable information applicable to numerous situations, such as inflammatory processes. For instance, the ECW to TBW ratio of ≥0.4 could predict treatment failure among patients with advanced lung cancer or poorer relapse-free survival in patients with correctional cancer [14,16]. In addition, TBW and ICW were correlated with cardiovascular disease [19]. Hence, monitoring the changes of body water compartments are extremely important.

Methods to measure body water in humans includes isotope dilution, air displacement plethysmography (ADP), dual-energy X-ray absorptiometry (DEXA), and bioelectrical impedance analysis (BIA) [20]. The isotope dilution is the golden method, but with highly expensive cost and difficulty in operation. While it is worth mentioning that BIA has been proved with high validity and accuracy to determine total body water content [21,22,23,24,25,26,27], even for acute water supplementation of 120 min, significant changes in ICW and ECW were observed using BIA among young adults [28]. Therefore, BIA has been able to fill the aforementioned scientific gaps in measuring body water content in a large sample of people.

In China, only a few studies have accurately assessed the changes of TBW and its compartments among adults [24,29], not to say among children or adolescents. One study reported the results of water contents of young healthy adults aged 18–23 years, mainly showing that the TBW and the compartments differed from that of adults with the same age in other countries [24]. Another research study demonstrated that TBW decreased among healthy adults after 36 h water restriction [29]. However, the characteristics of body water content have not been fully evaluated and studied among children and adolescents, and comprehensive large-sample studies are lacking. As a result, it is challenging to gauge the growth trajectory and related health perils of children but more data are needed to understand this issue.

To address this deficiency, we devised the present investigation to examine the water content of the body using BIA among Chinese children and adolescents aged 6–17 years in free-living conditions, including factors that affect total body water, including gender, age and, nutrition status. Our study should provide basic information of body water and its constituent compartments of children in China, and could give rise to a means of education on fluid intake that is based on science for children in order to promote their healthy development.

## 2. Methods

### 2.1. Subjects

A cross-sectional study was formulated, which consisted of 196 individuals in each age group (28 strata,14 provinces, 7 regions, 2 urban/rural samples, half males and half females). We employed a multi-stage stratified randomized cluster sampling technique to compute the sample size. In total, 65,856 children aged 6–17 years were required for the study. The inclusion criteria were as follows: age between 6 to 17 years; healthy children without diseases as diagnosed by hospital; having stayed in the survey location for longer than 6 months; and having completed a written informed consent form. Exclusion criteria were set as follows: individuals with either acute or chronic ailments, including acute respiratory tract infection, acute gastroenteritis, and congenital heart defects, were not included in the study [30].

### 2.2. Anthropometric Measurements

The height, weight, and body composition were assessed in the morning with participants having fasted overnight. The height of the participants was gauged with an accuracy of up to 0.1 cm, while the weight was measured with a precision of up to 0.1 kg. Furthermore, this was done when the participants were dressed in light clothing and barefoot—digital weight scales (GMCS-I electronic scale; Jianmin, Beijing, China) Body mass index (BMI) was defined as the ratio of weight (kg) to the square of height (m^2^).

Body water contents, including TBW, ICW and ECW, FFM were determined by a bioelectrical impedance analyzer (BIA) (Inbody 770; Inbody; Seoul, Korea) with trained investigators, in which participants were also asked to be in light clothing and barefoot. In accordance with the four-component model theory of body composition, the instrument is equipped with 8-point contact electrodes, specifically two thumb electrodes, two palm electrodes, two sole electrodes, and two heel electrodes. It functions by measuring 30 impedance values across 5 body segments, namely the left and right upper limbs, the trunk, and the lower limbs, and this is achieved at 6 distinct frequencies: 1, 5, 50, 250, 500, and 1000 kHz. By employing diverse high-frequency and low-frequency settings, it can measure the intracellular and extracellular water, respectively, thereby enabling an accurate analysis of the overall water content [31].

From the body water compartments, ratios were calculated for further analyses, namely TBW/BW (TBW to BW ratio), ICW/TBW (ICW to TBW ratio), ECW/TBW (ECW to TBW ratio), ECW/ICW (ECW to ICW ratio), and TBW/FFM (TBW to FFM ratio).

### 2.3. Statistics

The statistical analysis of the current study was carried out using the SAS 9.2 software (SAS Institute Inc., Cary, NC, USA). Age was categorized in four quartiles: age 6–8 years, age 9–11 years, age 12–14 years, and age 15–17 years. BMI was classified according to the health industry standards in China—Screening Standard for Malnutrition in School-age Children and Adolescents: WS/T 456-2014 [32]; Screening for overweight and obesity among school-age children and adolescents: WS/T 586-2018 [33]—into underweight, normal weight, overweight, and obesity. Data were presented in two different ways depending on their distribution. If the data were normally distributed, they were shown as the mean ± standard deviation; if not, they were presented as the median and interquartile ranges (IQR = 25th–75th percentile). Differences of variables in normal distribution including the proportions TBW/BW, ICW/TBW, ECW/TBW, ECW/ICW, and TBW/FFM among the four age groups, and three nutrition status groups were used for one-way ANOVA; the Scheffe’s method for multiple tests was employed, and the alpha level was set at 0.05, both in boys and girls. Differences of variables in skewness distribution such as the TBW, ICW, ECW, and FFM among the four age groups and three nutrition status groups were assessed by the Kruskal–Wallis H test, both in boys and girls; the alpha level was set at 0.008 (0.05/6 comparisons) and 0.017 (0.05/3 comparisons) for multiple tests. Mann–Whitney U test and Student’s t-test were used to compare differences between groups of sexes with the same age. Statistical significance was set at *p* < 0.05.

## 3. Results

The study included 59,643 participants, including 30,103 boys and 29,540 girls. About 6213 participants quitted the study. The participant characteristics are displayed in Table 1. The height, weight, and BMI differed significantly between boys and girls (all *p* < 0.05). The flow chart is shown in Figure 1.

The TBW, ICW, and ECW of the participants were 24.4 kg, 15.0 kg, and 9.4 kg, respectively; the proportions of ICW/TBW, ECW/TBW, TBW/FFM, and TBW/BW were 61.6%, 38.4%, 73.1%, and 56.1%, respectively. The characteristics of body water contents of subjects with different ages are illustrated in Table 2. As the children became older, the TBW, ICW, ECW, ICW/TBW, and TBW/FFM increased year by year, while, the ECW/TBW decreased simultaneously with age. For boys, the TBW, ICW, and ECW increased with age (all *p* < 0.001), for which the TBW and ICW increased from 15.3 kg and 9.4 kg in age 6–7 years to 37.5 kg and 23.4 kg in age 16–17 years (*p* < 0.05), respectively. For girls, the same trend was observed (all *p* < 0.001). The values of TBW and ICW increased from 14.3 kg and 8.8 kg for subjects of age 6–7 years to 27.6 kg and 17.0 kg for subjects of age 16–17 years, respectively (all *p* < 0.001). Furthermore, after the age of 11 years, the TBW, ICW, and ECW increased rapidly, observed both in boys and girls.

Proportions TBW/BW, ICW/TBW, ECW/TBW, and TBW/FFM were different among the age groups, for both genders of children (all *p* < 0.001). As for ICW/TBW, the values for boys were all higher than girls at the same age (*p* < 0.05), except for age 6 years. The rates of the changes in ICW/TBW surpassed those of the girls, with the girls reaching their peak at the age of 14 years; while the boys increased after reaching their peak at 8 to 12 years of age, ranging from 61.6% at age 6–7 years to 62.3% at the age 16–17 years. As for ECW/TBW, the trends of boys and girls all decreased with age, and the values of each age were all lower in boys than those of girls (all *p* < 0.05). Moreover, the ECW/TBW decreased after reaching peaks at 8 to 12 years of age, but that of girls did not decrease from 14 to 17 years of age. However, the trends of TBW/FFM with age differed between boys and girls, but the values of boys exceeded those of the girls (all *p* < 0.05). The TBW/FFM of boys did not significantly change at the ages of 8–10 years and 12–15 years, but rapidly decreased after the age of 15 years (*p* < 0.05). However, the peaks of TBW/FFM in girls were at the ages of 11–12 years and 15–17 years (*p* < 0.05), as depicted in Figure 2.

The body water compartments of subjects with different BMI are given in Table 3 and Figure 3. All BMI groups showed differences in TBW, ICW, ECW, and FFM, with the highest contents of TBW, ICW, ECW, and FFM in the overweight and obese group, among both boys and girls (all *p* < 0.001). Additionally, the ICW/TBW, ECW/TBW, TBW/BW, ECW/ICW, and TBW/FFM differed significantly in the three BMI groups (all *p* < 0.001), but with different patterns. The children in the obesity group showed clearly lower levels of TBW/BW at all ages (all *p* < 0.001); with boys having higher proportions than those of girls with the same age (all *p* < 0.001). Moreover, TBW/BW demonstrated a decrease first and increased at the age of 12 years, but the decrease was marked in the overweight and obese and underweight group; the normal group showed a smaller decrease. Otherwise, the TBW/BW, ECW/TBW, and ECW/ICW showed decreases among the BMI groups, and the ICW/TBW increased among the groups (all *p* < 0.001). Furthermore, lower BMI groups presented lower levels of TBW/FFM, simultaneously; the higher BMI groups showed higher levels of TBW/FFM, both in girls and boys (all *p* < 0.001). The normal group and the overweight group showed no differences in the variance of TBW/FFM (*p >* 0.05), but with the highest one in the obese group and the lowest in the underweight group (all *p* < 0.001), as shown in Figure 3.

## 4. Discussion

As far we know, the current study was the maiden attempt to explore the differences in body water content among children within the age range of 6–17 years in China.

As expected, in our study, the TBW, ICW, ECW increased simultaneously with age, and boys had higher indices than girls of the same age, which was consistent with other studies in Poland and Spain [34,35]. In one study, which examined ballet school aged 14 ± 2 and elementary school students aged 10 ± 2 in Poland, the results showed that the TBW increased from elementary school children to ballet high school children [35]. These results further demonstrate that as children grow older, it could bring about changes in their body water compartments. Interestingly, even among children of the same age, the numbers of TBW, ICW, and ECW of children from the United States, Spain, and Portugal were different compared to that of the present study [17,36,37]. The contents of TBW, ICW, and ECW among children aged 6–17 years in Portugal (26.7 kg, 15.1 kg, and 11.6 kg) were all higher than the results of ours (24.4 kg, 15.0 kg, and 9.4 kg), respectively [17]. Differences in the methods measuring the body water contents or race might account for the disputable findings of the research studies. The current study used BIA to determine the TBW and its compartments, whereas, in the other study in Spain, the TBW was assessed by isotope dilution. Children apart, in adults similar differences of body water compartments were found among different countries using the BIA to evaluate the body water contents [24,38,39]. Notwithstanding, the TBW differed among children of the same age with racial/ethnic groups using the same methods in evaluating body composition [40]. In this study, the TBW of Black children aged between 4 and 6 years, 7 and 9 years, 10 and 12 years, as well as 13 and 15 years, which were measured by deuterium dilution, were all higher than that of White and Hispanic children.

As previously mentioned, the values of TBW, ICW, ECW, TBW/BW, ICW/TBW, ECW/TBW, ECW/ICW, and TBW/FFM of both genders of children of different ages varied significantly. In addition, our research revealed that, in comparison to girls, boys exhibited higher values for TBW, ICW, ECW, and the ratio of TBW to fat-free mass (TBW/FFM). Moreover, these values demonstrated a steady upward trend as age advanced. The changes of TBW/BW with aging have been documented in other studies among children and adults [37,41]. Interestingly, in the current study, the percentages of TBW/BW presented a different change trend among age groups, ranging from 50.8% to 59.1%. Further on, TBW/BW decreased from 58.4% to 55.4%, reaching then the age of 9–11 years, and then a slight increase to 59.1% was observed in boys between 15 and 17 years. While girls exhibited a continuously decreasing trend in their relative values from 58.5% to 50.8%; as previously reported [42]. Moreover, we observed that compared with girls, boys had higher TBW/BW, except at the age of 9–11 years old. This may be attributed to the relatively higher increased body weight and muscle mass in boys than girls [43].

In the current research, the proportion of ECW/TBW increased with age, and in all of the age groups under consideration, girls had a higher value than boys (*p* < 0.001). Similar results were found in another large sample study of adults [22]. As for ECW/ICW, according to one research study, women presented a greater ECW/ICW ratio [44], which was consistent with our results that in all age group, the ECW/ICW was higher in girls and increased with age, but remained stable at 0.38. In another study conducted among 1992 healthy individuals aged ≥ 15 years in Japan, the findings indicated that as age advanced, the ratio of ECW to ICW grew because the decline in ICW content was more pronounced compared to that of ECW [45]. Nevertheless, controversial results were demonstrated in a study conducted among children aging in the United States, in that no significant differences of ECW/ICW were found in age groups, nor in boys or girls [46]. The results could be explained by differences of race. Similar trends were observed when comparing with Black, White, and Hispanic children in the United States [40]. The ratio was stable at 0.730–0.732 in our study, lower than that of adults, which may be explained by the higher proportions of water and the differences of mineral and protein when comparing children with adults [47], meaning that the children were in optimal hydration status in our study. Meanwhile, the values of TBW/FFM were lower than the results obtained from a study carried out among children within the age range of 6–16 years in the United States, in which the proportions of TBW/FFM were 74.2% [48]. The methods to assess and calculate the TBW/FFM result in the differences in the outcomes. Hence, alterations in total body water could have an impact on overall health, which confirms the importance of the balance of the body water compartments.

It demonstrated that as children increased in BMI in all age groups from underweight to obesity, the values of TBW made a significantly lower percentage to BW, both in girls and boys. This might be due to the fact that the excessive body weight (BW) stemmed from a relatively high proportion of body fat as compared to the proportions of FFM and TBW in relation to BW [49], even though the TBW and FFM increased with BMI in all age groups. Because the contributions of water to body fat and FFM were ~10% and 75% [50], overweight and obese individuals had lower TBW/BW, which has been demonstrated among adults [34]. Additionally, in our research, a gender disparity was observed in this parameter; specifically, at the same BMI, boys were found to possess more water per kilogram of body weight compared to girls, which is consistent with the results of the studies implemented among children, adolescents, and healthy adults [51,52,53]. However, previous studies revealed that participants of overweight were hypohydrated when compared with those of normal weight [54,55,56]. Unexpected slight changes of TBW/BW were found in children with normal weight, and large fluctuations were observed among those with obesity. Thereby, lower values of TBW/BW among overweight and obesity children are noteworthy, as there is a higher proportion of children with obesity around the world. More researche studies are needed to evaluate the effects of lower proportions of TBW to BW on the health of children with obesity, who were shown to have less water and a higher risk to be hypohydrated [57].

There have been few data evaluating the distribution of human body water between ECW and ICW among children. Our study showed that the ECW/TBW increased and ICW/TBW decreased with BMI, and was higher in girls than boys over the whole BMI range. Moreover, at similar BMI, girls had a higher proportion of body fat than boys, therefore, higher ECW/TBW as expected [22]. It was proved that the indicator of cellular hydration, TBW/FFM, was not only impacted by age but also impacted by nutrition status, in other words, the BMI [55]. The increasing values of TBW/FFM were obtained in girls from normal weight to overweight and obesity throughout the age range, but only among boys at the age of 6 years to 11 years, in our study, which is similar to the study implemented in a small sample of 28 obese children and 22 non-obese children [52]. The mechanism that expanded ECW may bring about the shifts in the constitution of FFM associated with obesity [58]. Furthermore, another study suggested that the higher content of water and mineral and relatively lower percentage of protein in obese children may be attributed to the higher hydration status [51]. But there may be more mechanisms leading to the alterations of TBW/FFM, which are also in need of more attention.

Our research possessed certain advantages and disadvantages. First, in China, our research was primarily to assess the influential factors including age, sex, and nutrition status of body water among adolescents including males and females in free-living conditions and it was of a substantial sample size over a wide range of age and BMI. Although our study had certain notable strengths as previously described, it was not without its limitations. The research was specifically conducted among children, which means that when extending the study’s conclusions to other age brackets, caution must be exercised. Going forward, further investigations will explore the broader applicability of our findings to a more diverse demographic, encompassing adults as well as the elderly.

## 5. Conclusions

The body water content and the changes of children aged 6–17 years in China varied according to their age, sex, and BMI. Overweight and obese individuals had significantly lower TBW/BW and higher TBW/FFM than those with normal weight. Hence, in order to maintain stability of the body water content, science-based education of fluid intake for children is needed in the light of the distinct traits exhibited by children, particularly those who are overweight.

## Figures and Tables

**Figure 1 nutrients-17-00364-f001:**
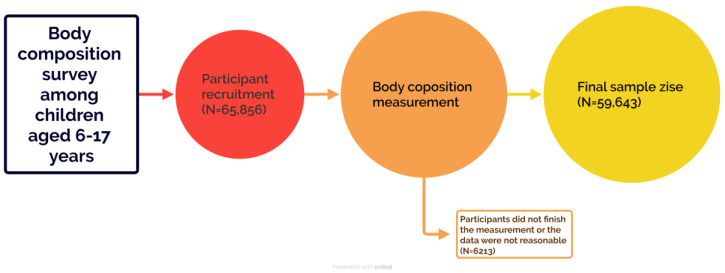
The flow chart.

**Figure 2 nutrients-17-00364-f002:**
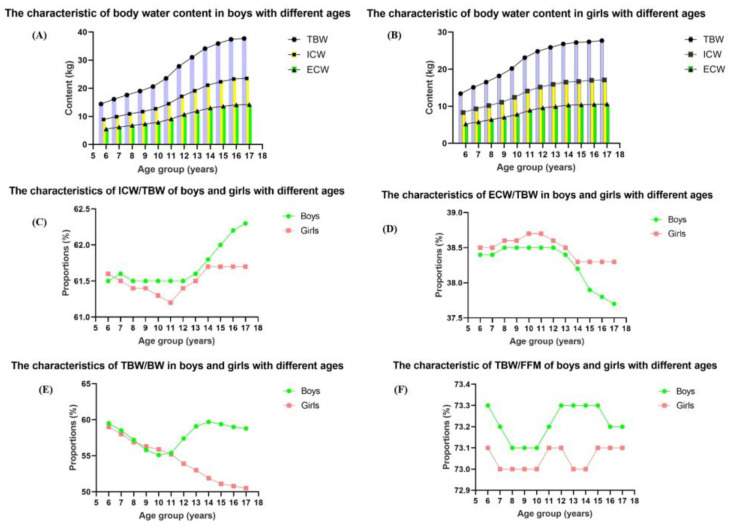
The body water content of boys and girls with different age groups. (**A**): The characteristic of body water content in boys with different ages; (**B**): The characteristic of body water content in girls with different ages; (**C**): The characteristic of ICW/TBW of boys and girls with different ages; (**D**): The characteristic of ECW/TBW of boys and girls with different ages; (**E**): The characteristic of TBW/BW of boys and girls with different ages; (**F**): The characteristic of TBW/FFM of boys and girls with different ages.

**Figure 3 nutrients-17-00364-f003:**
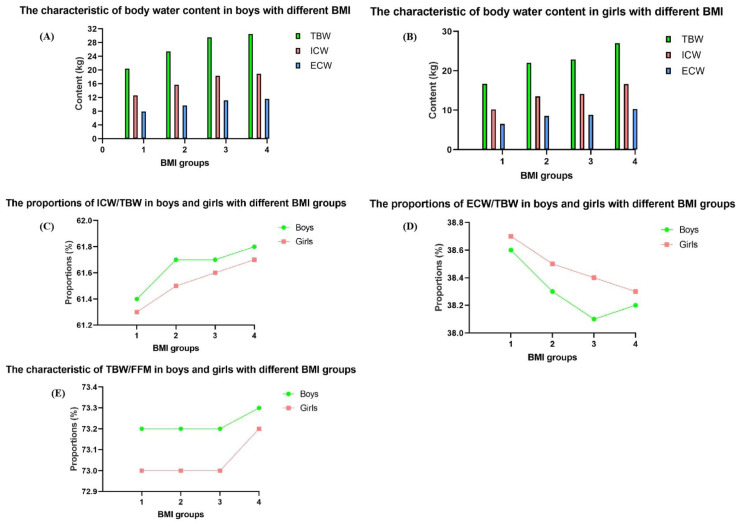
The body water content of boys and girls with different BMI groups. 1: Underweight group; 2: Normal weight group; 3: Overweight group; 4: Obese group; (**A**): The characteristic of body water content in boys with different BMI; (**B**): The characteristic of body water content in girls with different BMI; (**C**): The proportions of ICW/TBW in boys and girls with different BMI groups; (**D**): The proportions of ECW/TBW in boys and girls with different BMI groups; (**E**): The characteristic of TBW/FFM in boys and girls with different BMI groups.

**Table 1 nutrients-17-00364-t001:** Participant characteristics.

	Total (*n* = 59,643)	Boys (*n* = 30,103)	Girls (*n* = 29,540)
Age (y)	11.6 ± 3.4	11.6 ± 3.4	11.6 ± 3.4
Height (cm)	149.1 ± 17.5	151.2 ± 19.1	146.9 ± 15.4
Weight (kg)	44.4 ± 17.4	46.7 ± 19.1	42.4 ± 14.9
BMI (Kg/m^2^)	19.3 ± 4.7	19.5 ± 4.8	19.0 ± 4.4

Note: Values are shown as the mean ± standard deviation (SD). BMI: body mass index.

**Table 2 nutrients-17-00364-t002:** The body water content of participants of different age groups.

	Total	Age 6–8 Years	Age 9–11 Years	Age 12–14 Years	Age 15–17 Years	*p*
*Boys (n)*	30,103	7271	7580	7449	7803	
TBW (kg)	24.9 (17.5, 34.7)	15.4 (13.9, 17.3) *^†abc^	20.3 (17.9, 23.4) ^†de^	30.8 (26.2, 35.3) ^†f^	37.2 (33.6, 41.1) ^†^	<0.001
ICW (kg)	15.3 (10.7, 21.5)	9.5 (8.5, 10.7) *^†abc^	12.5 (11.0, 14.4) ^†de^	19.0 (16.1, 21.8) ^†f^	23.3 (20.9, 25.6) ^†^	<0.001
ECW (kg)	9.6 (6.7, 13.2)	5.9 (5.3, 6.6) *^†abc^	7.8 (6.9, 9.0) ^†de^	11.8 (10.1, 13.5) ^†f^	14.0 (12.7, 15.5) ^†^	<0.001
FFM (kg)	33.9 (23.9, 47.3)	21.1 (18.9, 23.6) *^†abc^	27.8 (24.5, 31.9) ^†de^	41.9 (35.7, 48.1) ^†f^	50.8 (45.9, 56.2) ^†^	<0.001
TBW/BW (%)	57.9 ± 6.7	58.4 ± 6.3 *^†abc^	55.4 ± 7.0 ^†de^	58.7 ± 6.7 ^†f^	59.1 ± 5.9 ^†^	<0.001
ICW/TBW (%)	61.7 ± 6.5	61.5 ± 6.5 *^†abc^	61.5 ± 4.9 ^†de^	61.6 ± 6.0 ^†f^	62.2 ± 6.0 ^†^	<0.001
ECW/TBW (%)	38.3 ± 6.3	38.4 ± 5.8 *^†abc^	38.5 ± 4.9 ^†de^	38.4 ± 6.0 ^†f^	37.8 ± 6.0 ^†^	<0.001
ECW/ICW (%)	62.0 ± 1.6	62.5 ± 1.3 *^†abc^	62.6 ± 1.3 ^†de^	62.3 ± 1.6 ^†f^	60.8 ± 1.6 ^†^	<0.001
TBW/FFM (%)	73.2 ± 5.3	73.2 ± 6.3 *^†abc^	73.2 ± 7.6 ^†de^	73.3 ± 2.9 ^†f^	73.2 ± 2.9 ^†^	<0.001
*Girls (n)*	29,540	4731	9948	7306	7555	
TBW (kg)	23.1 (16.5, 27.0)	13.7 (12.5, 15.1) ^#abc^	18.8 (16.1, 22.3) ^de^	25.6 (23.4, 28.1) ^f^	27.3 (25.2, 29.8)	<0.001
ICW (kg)	14.2 (10.2, 16.7)	8.4 (7.7, 9.3) ^#abc^	11.5 (9.9, 13.6) ^de^	15.7 (14.4, 17.3) ^f^	16.9 (15.6, 18.4)	<0.001
ECW (kg)	8.9 (6.4, 10.4)	5.3 (4.8, 5.8) ^#abc^	7.3 (6.2, 8.6) ^de^	9.8 (9.0, 10.8) ^f^	10.5 (9.7, 11.4)	<0.001
FFM (kg)	31.6 (22.6, 37.0)	18.8 (17.2, 20.6) ^#abc^	25.7 (22.1, 30.5) ^de^	35.0 (32.1, 38.4) ^f^	37.4 (34.6, 40.7)	<0.001
TBW/BW (%)	54.3 ± 6.0	58.5 ± 5.5 ^#abc^	56.0 ± 5.8 ^de^	52.9 ± 5.3 ^f^	50.8 ± 4.8	<0.001
ICW/TBW (%)	61.5 ± 6.3	61.5 ± 8.8 ^#ac^	61.3 ± 5.4 ^de^	61.5 ± 5.4 ^f^	61.7 ± 5.4	<0.001
ECW/TBW (%)	38.5 ± 5.7	38.5 ± 6.6 ^#abc^	38.7 ± 5.2 ^de^	38.5 ± 5.4 ^f^	38.3 ± 5.4	<0.001
ECW/ICW (%)	62.6 ± 1.4	62.6 ± 1.3 ^#ac^	63.0 ± 1.3 ^de^	62.5 ± 1.4 ^f^	62.0 ± 1.4	<0.001
TBW/FFM (%)	70.0 ± 4.8	73.0 ± 8.4 ^#abc^	73.0 ± 5.1 ^de^	73.0 ± 2.6 ^f^	73.1 ± 2.6	<0.001

Note: Values are shown as the mean ± standard deviation (SD) and medians (IQR). TBW: total body water; ICW: intracellular water; ECW: extracellular water; FFM: far free mass; ICW/TBW: ICW to TBW ratio; ECW/TBW: ECW to TBW ratio; ECW/ICW: ECW to ICW ratio; TBW/BW: TBW to BW ratio; TBW/FFM: TBW to FFM ratio. One-way ANOVA was used to comparing the differences of variables including the proportions TBW/BW, ICW/TBW, ECW/TBW, ECW/ICW, and TBW/FFM among the four age groups; the Scheffe’s method for multiple tests was used, both in boys and girls. Differences of the TBW, ICW, ECW, and FFM among the four age groups were assessed by Kruskal–Wallis H test, and also used for the multiple tests, both in boys and girls. Student’s t-test was used to compare differences in the proportions TBW/BW, ICW/TBW, ECW/TBW, ECW/ICW, and TBW/FFM between boys and girls with the same age group; Mann–Whitney U test was used to compare differences in TBW, ICW, ECW, and FFM between boys and girls with the same age group. *: Significant differences were found between different age groups among boys (*p* < 0.05); ^#^: significant differences were found between different age groups among girls (*p* < 0.05); ^†^: significant differences were found between boys and girls in the same age group (*p* < 0.05). ^a^: There were significant differences between age 6–8 years group and age 9–11 years group (*p* < 0.05 for normal data and *p* < 0.008 for data in skewness distribution); ^b^: there were significant differences between age 6–8 years group and age 12–14 years group (*p* < 0.05 for normal data and *p* < 0.008 for data in skewness distribution); ^c^: there were significant differences between age 6–8 years group and age 15–17 years group (*p* < 0.05 for normal data and *p* < 0.008 for data in skewness distribution); ^d^: there were significant differences between age 9–11 years group and age 12–14 years group (*p* < 0.05 for normal data and *p* < 0.008 for data in skewness distribution); ^e^: there were significant differences between age 9–11 years group and age 15–17 years group (*p* < 0.05 for normal data and *p* < 0.008 for data in skewness distribution); ^f^: there were significant differences between age 12–14 years group and age 15–17 years group (*p* < 0.05 for normal data and *p* < 0.008 for data in skewness distribution).

**Table 3 nutrients-17-00364-t003:** The body water content of participants in different age groups of nutrition status.

	Underweight					Normal Weight					Overweight and Obese				
	Total	Age 6–8 Years	Age 9–11 Years	Age 12–14 Years	Age 15–17 Years	Total	Age 6–8 Years	Age 9–11 Years	Age 12–14 Years	Age 15–17 Years	Total	Age 6–8 Years	Age 9–11 Years	Age 12–14 Years	Age 15–17 Years
*Boys (n)*	2417	482	556	610	769	15,163	4513	4371	4633	1646	7442	2276	2653	2206	307
TBW (kg)	18.1 (14.2, 27.3)	13.5 (12.5, 14.6) *^gh^	16.9 (15.3, 18.3) ^gh^	22.1 (19.1, 26.4) ^gh^	29.5 (13.8, 32.8) ^gh^	20.5 (16.1, 30.1)	14.8 (13.5, 16.2) *^l^	19.2 (17.4, 21.5) ^l^	29.8 (25.8, 33.5) ^l^	35.2 (32.8, 38.0) ^l^	24.4 (19.4, 33.6)	17.6 (15.8, 20.1) *	23.4 (21.0, 26.7)	35.8 (31.0, 40.2)	39.8 (37.4, 42.5)
ICW (kg)	11.1 (8.7, 16.8)	8.3 (7.7, 9.0) *^gh^	10.3 (9.4, 11.3) ^gh^	13.6 (11.7, 16.2) ^gh^	18.3 (8.5, 20.3) ^gh^	12.6 (9.9, 18.6)	9.1 (8.3, 10.0) *^l^	11.8 (10.7, 13.2) ^l^	18.4 (15.9, 20.7) ^l^	21.8 (20.3, 23.7) ^l^	15.0 (11.9, 20.7)	10.9 (9.7, 12.4) *	14.4 (12.9, 16.4)	22.1 (19.1, 24.8)	24.9 (23.3, 26.4)
ECW (kg)	7.0 (5.5, 10.5)	5.2 (4.8, 5.6) *^gh^	6.5 (5.9, 7.1) ^gh^	8.6 (7.4, 10.3) ^gh^	11.3 (5.3, 12.5) ^gh^	7.9 (6.2, 11.5)	5.7 (5.2, 6.2) *^l^	7.4 (6.7, 8.3) ^l^	11.5 (10.0, 12.8) ^l^	13.3 (12.4, 14.4) ^l^	9.4 (7.4, 12.9)	6.7 (6.1, 7.7) *	9.0 (8.0, 10.3)	13.6 (11.9, 15.2)	15.0 (14.0, 16.0)
FFM (kg)	24.7 (19.5, 37.2)	18.5 (17.2, 20.1) *^gh^	23.1 (21.0, 25.2) ^gh^	30.2 (26.1, 35.9) ^gh^	40.2 (18.8, 44.6) ^gh^	28.1 (22.0, 41.1)	20.3 (18.5, 22.2) *^l^	26.3 (23.8, 29.3) ^l^	40.6 (35.3, 45.7) ^l^	48.0 (44.8, 51.8) ^l^	33.3 (26.4, 45.9)	24.0 (21.5, 27.4) *	32.0 (28.6, 36.4)	48.8 (42.4, 54.9)	54.5 (51.2, 58.0)
TBW/BW (%)	64.0 ± 3.2	65.2 ± 2.7 *^gh^	63.5 ± 2.9 ^gh^	64.0 ± 3.4 ^gh^	63.6 ± 3.4 ^gh^	60.7 ± 4.0	61.1 ± 3.4 *^l^	58.9 ± 4.1 ^l^	61.6 ± 4.2 ^l^	61.9 ± 3.5 ^l^	50.3 ± 5.5	51.4 ± 5.5 *	48.0 ± 4.8	51.1 ± 5.5	56.2 ± 3.6
ICW/TBW (%)	61.4 ± 0.6	61.4 ± 0.6 *^gh^	61.3 ± 0.5 ^gh^	61.3 ± 0.6 ^gh^	61.7 ± 0.6 ^gh^	61.6 ± 6.1	61.5 ± 0.7 *^l^	61.5 ± 0.5 ^l^	61.6 ± 0.6 ^l^	62.1 ± 0.6 ^l^	61.7 ± 0.6	61.6 ± 0.5 *	61.6 ± 0.5	61.8 ± 0.6	62.3 ± 0.6
ECW/TBW (%)	38.6 ± 0.6	38.6 ± 0.5 *^gh^	38.7 ± 0.5 ^gh^	38.7 ± 0.6 ^gh^	38.3 ± 0.6 ^gh^	38.4 ± 5.8	38.5 ± 0.6 *^l^	38.6 ± 0.5	38.4 ± 0.6 ^l^	37.9 ± 0.6 ^l^	38.3 ± 0.6	38.4 ± 0.5 *	38.4 ± 0.5	38.2 ± 0.6	37.7 ± 0.6
ECW/ICW (%)	62.8 ± 1.5	62.9 ± 1.4 *^gh^	63.2 ± 1.3 ^gh^	63.2 ± 1.5 ^gh^	62.2 ± 1.5 ^gh^	62.4 ± 1.5	62.6 ± 1.3 *^l^	62.7 ± 1.3 ^l^	62.4 ± 1.5 ^l^	61.1 ± 1.5 ^l^	62.1 ± 1.4	62.2 ± 1.3 *	62.3 ± 1.3	61.9 ± 1.6	60.5 ± 1.5
TBW/FFM (%)	73.2 ± 0.5	72.9 ± 0.5 *^gh^	72.9 ± 0.3 ^gh^	73.3 ± 0.4 ^gh^	73.5 ± 0.5 ^gh^	73.2 ± 4.6	73.2 ± 0.7 *^l^	73.1 ± 0.3 ^l^	73.3 ± 0.3 ^l^	73.3 ± 0.2 ^l^	73.3 ± 0.8	73.4 ± 0.4 *	73.3 ± 1.1	73.2 ± 0.3	73.1 ± 0.2
*Girls (n)*	1650	285	614	288	463	20,375	3952	5482	5505	5436	4229	494	1113	966	1656
TBW (kg)	15.3 (12.9, 20.5)	12.3 (11.5, 13.1) ^#gh^	14.9 (13.6, 16.7) ^gh^	19.9 (17.2, 22.0) ^gh^	21.8 (11.7, 24.5) ^gh^	23.1 (16.9, 26.4)	13.6 (12.5, 14.7) ^#l^	19.3 (17.1, 22.0) ^l^	25.1 (23.1, 27.0) ^l^	26.8 (25.1, 28.7) ^l^	28.9 (24.7, 31.9)	17.0 (15.4, 20.0) ^#^	24.4 (20.6, 28.2)	30.4 (28.1, 32.9)	30.8 (28.5, 33.1)
ICW (kg)	9.4 (7.9, 12.5)	7.5 (7.1, 8.1) ^#gh^	9.1 (8.4, 10.2) ^gh^	12.2 (10.5, 13.5) ^gh^	13.4 (7.3, 15.1) ^gh^	14.2 (10.4, 16.2)	8.3 (7.7, 9.1) ^#l^	11.9 (10.5, 13.5) ^l^	15.4 (14.2, 16.6) ^l^	16.6 (15.5, 17.7) ^l^	17.8 (15.2, 19.7)	10.5 (9.5, 12.3) ^#^	15.0 (12.7, 17.3)	18.8 (17.4, 20.3)	19.0 (17.6, 20.5)
ECW (kg)	5.9 (5.0, 7.9)	4.8 (4.4, 5.1) ^#gh^	5.8 (5.3, 6.5) ^gh^	7.7 (6.7, 8.6) ^gh^	8.5 (4.5, 9.4) ^gh^	8.9 (6.5, 10.1)	5.2 (4.8, 5.7) ^#l^	7.5 (6.6, 8.5) ^l^	9.6 (8.9, 10.4) ^l^	10.3 (9.6, 11.0) ^l^	11.0 (9.4, 12.2)	6.5 (5.9, 7.7) ^#^	9.4 (7.9, 10.9)	11.6 (10.8, 12.6)	11.7 (10.9, 12.6)
FFM (kg)	21.0 (17.7, 27.9)	16.9 (15.8, 18.0) ^#gh^	20.5 (18.8,22.9) ^gh^	27.2 (23.5, 30.1) ^gh^	29.9 (16.0, 33.5) ^gh^	31.7 (23.2, 36.1)	18.6 (17.2, 20.2) ^#l^	26.5 (23.5, 30.1) ^l^	34.3 (31.7, 36.9) ^l^	36.7 (34.3, 39.3) ^l^	39.5 (33.7, 43.7)	23.3 (21.0, 27.2) ^#^	33.4 (28.2, 38.5)	41.6 (38.5, 45.0)	42.1 (39.0, 45.2)
TBW/BW (%)	61.8 ± 4.0	65.0 ± 2.9 ^#gh^	62.8 ± 3.2 ^gh^	61.2 ± 3.3 ^gh^	58.7 ± 3.8 ^gh^	55.5 ± 4.7	59.2 ± 4.1 ^#l^	57.5 ± 4.0 ^l^	54.3 ± 4.0 ^l^	51.9 ± 3.4 ^l^	45.9 ± 4.0	48.6 ± 5.0 ^#^	46.4 ± 4.3	45.1 ± 3.3	45.2 ± 3.5
ICW/TBW (%)	61.3 ± 0.6	61.3 ± 1.0 ^#gh^	61.2 ± 0.5 ^gh^	61.1 ± 0.5 ^gh^	61.4 ± 0.6 ^gh^	61.5 ± 0.6	61.5 ± 0.9 ^#l^	61.3 ± 0.5 ^l^	61.5 ± 0.5 ^l^	61.7 ± 0.5 ^l^	61.7 ± 0.6	61.7 ± 0.5 ^#^	61.5 ± 0.6	61.8 ± 0.5	61.9 ± 0.5
ECW/TBW (%)	38.7 ± 0.6	38.6 ± 0.7 ^#gh^	38.8 ± 0.5 ^gh^	38.9 ± 0.5 ^gh^	38.6 ± 0.6 ^gh^	38.5 ± 0.6	38.5 ± 0.7 ^#l^	38.7 ± 0.5 ^l^	38.5 ± 0.5 ^l^	38.3 ± 0.5 ^l^	38.3 ± 0.6	38.4 ± 0.5 ^#^	38.5 ± 0.6	38.2 ± 0.5	38.1 ± 0.5
ECW/ICW (%)	63.2 ± 1.4	62.9 ± 1.4 ^#gh^	63.5 ± 1.4 ^gh^	63.6 ± 1.3 ^gh^	62.9 ± 1.5 ^gh^	62.6 ± 1.4	62.6 ± 1.3 ^#l^	63.2 ± 1.3 ^l^	62.6 ± 1.4 ^l^	62.0 ± 1.4 ^l^	62.0 ± 1.4	62.2 ± 1.2 ^#^	62.3 ± 1.3	61.9 ± 1.4	61.6 ± 1.4
TBW/FFM (%)	73.0 ± 0.7	72.9 ± 1.1 ^#gh^	72.8 ± 0.4 ^gh^	73.0 ± 0.3 ^gh^	73.3 ± 0.5 ^gh^	73.0 ± 0.5	73.0 ± 0.8 ^#l^	73.0 ± 0.5 ^l^	73.0 ± 0.3 ^l^	73.0 ± 0.2 ^l^	73.2 ± 0.3	73.4 ± 0.4 ^#^	73.3 ± 0.3	73.1 ± 0.2	73.1 ± 0.2

Note: Values are shown as the mean ± standard deviation (SD) and median and inter quartile range (IQR = 25th–75th percentile). TBW: total body water; ICW: intracellular water; ECW: extracellular water; FFM: far free mass; ICW/TBW: ICW to TBW ratio; ECW/TBW: ECW to TBW ratio; ECW/ICW: ECW to ICW ratio; TBW/BW: TBW to BW ratio; TBW/FFM: TBW to FFM ratio. One-way ANOVA was used to comparing the differences of variables including the proportions TBW/BW, ICW/TBW, ECW/TBW, ECW/ICW, and TBW/FFM among the three BMI groups; the Scheffe’s method for multiple tests was used; both in boys and girls. Kruskal–Wallis H test was used in comparing the differences of the TBW, ICW, ECW, and FFM among the three BMI groups and also was used for the multiple tests, both in boys and girls. *: Significant differences were found between groups among boys with same age in different BMI groups (*p* < 0.05); ^#^: significant differences were found between groups among girls with same age in different BMI groups (*p* < 0.05). ^g^: There were significant differences between underweight group and normal weight group with the same age (*p* < 0.05 for normal data and *p* < 0.017 for data in skewness distribution); ^h^: there were significant differences between underweight group and overweight and obese group (*p* < 0.05 for normal data and *p* < 0.017 for data in skewness distribution); ^l^: there were significant differences between normal weight group with overweight and obese group (*p* < 0.05 for normal data and *p* < 0.017 for data in skewness distribution).

## Data Availability

Data described in the manuscript will be made available upon request pending application and approval from the corresponding author.

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
