# Peer review of "Overweight and Obese Children Aged 6–17 Years in China Had Lower Level of Hydration Status: A Cross-Sectional Study"

_nutrients, 2025, doi:10.3390/nu17020364_

Round 1
Reviewer 1 Report
Comments and Suggestions for Authors
Zhang et al. conducted a cross-sectional study involving 59,643 Chinese children aged 6-17 years. The aim of the study was to evaluate the effect of age and BMI groups (underweight, normal, overweight, obesity) on various markers of total body water and hydration status in each sex group. More specifically, they assessed total body water (TBW), intracellular water (ICW) and extracellular water (ECW). The authors concluded that overweight and obese children had significantly lower TBW/BW and higher TBW/FFM (FFM=Fat Free Mass) than those of normal weight. Therefore, they recommend that to maintain the stability of body water content, it is necessary to educate children on fluid intake according to their different characteristics, especially in the case of overweight children.
Major comments.
The data analysis is erratic. The authors indicate that differences in groups of variables were compared using analysis of variance (line 7), which is not developed in the statistical analysis subsection, and worse, is not really done. In addition, the way in which the analyses are performed (summary of the variables in means and standard deviations) leads to the conclusion that all variables in all groups are normally distributed, which is not justified. These issues are detailed in the following points.
1. Table 2 is erratic for the following reasons:
a. First: the authors have summarized all markers of total body water and hydration status in means and standard deviations. We understand that the authors have adequately tested the normality assumptions. Do all markers really follow a normal distribution in all age groups? In such a case, the simultaneous comparison between the four groups should be done with a single hypothesis test which, given normality, should be the F-test corresponding to the one-way ANOVA.
b. If the F-test showed no significant differences, then pairwise comparisons are not appropriate. Otherwise, pairwise comparisons (six comparisons) should be made using a multiple comparisons test, for example, the Scheffee test. The use of the t-test is spurious (propagation of alpha error).
2. Table 3 is absolutely spurious. The authors intend to analyze the effect of age and BW (grouped in the three categories) on the markers of total body water and hydration status in means and standard deviations in each sex group. Given the normality of these markers, they should use an analysis of variance model (two-way ANOVA) with two factors of variance and interactions, which the authors have not done. Through this analysis, the authors should have tested for the following effects, namely: BW effect, age effect and the interaction between the two factors. Subsequently and when appropriate, the corresponding multiple comparisons should have been performed.
Minor comments
1. Improve the following expression: “Data was presented as mean ± standard deviation and median and quartile ranges, if the data was normal distributed or not, respectively”. This one would be better: “Data was presented as mean and standard deviation (SD) when data followed a normal distribution, or as median and interquartile range (IQR = 25th – 75th percentile) when distribution departed from normality”.
2. Table 1. Change: “The characteristics of participants” to “Participant characteristics”. Likewise, use the thousands separator. For example, change “59643” to “59,643”.
3. Figure 2 is inconsistent with the study design. For example, the authors have grouped the study subjects into four age groups, but in these figures age has been considered a continuous variable. In reality, a better analysis would have been to always use age on a continuous scale, if possible using at least one decimal place. This would have provided a broad age design that would have allowed the use of additive cubic spline models [1]. Such models are implemented in the mccv procedure corresponding to the R package [2]. The present figures are only a poor description of the data, as they do not provide whether or not the effects are significant.
4. In expresions: Proportions of TBW/BW”, “ICW/TBW”, “ECW/TBW” and “TBW/FFM” the “of” is incorrect. Change to: Proportions “TBW/BW”, “ICW/TBW”, “ECW/TBW” and “TBW/FFM”.
5. What is the style of the references used by the authors? It is certainly not Vancouver's.
This study seems to be a continuation of another study in which the first author participates [3]. The present one seems promising because of the large sample size, but the statistical analysis should be redone by an expert.
References
[1] Hastie T, Tibshirani R. Generalized additive models for medical research. Stat Methods Med Res. 1995 Sep;4(3):187-96.
[2] R Core Team (2022). R: A language and environment for statistical computing. R Foundation for Statistical Computing, Vienna, Austria. URL <https://www.R-project.org/>.
[3] Zhang N, Zhang J, Du S, He H, Yan X, Ma G. Association between the content of intracellular and extracellular fluid and the amount of water intake among Chinese college students. Nutr Metab (Lond). 2019 Sep 18;16:67. doi: 10.1186/s12986-019-0397-9. PMID: 31548843; PMCID: PMC6751809.
Author Response
Reviewer 1
Open Review
( ) I would not like to sign my review report
(x) I would like to sign my review report
Quality of English Language
(x) The quality of English does not limit my understanding of the research.
( ) The English could be improved to more clearly express the research.
|
|
Yes |
Can be improved |
Must be improved |
Not applicable |
Does the introduction provide sufficient background and include all relevant references? |
( ) |
(x) |
( ) |
( ) |
Is the research design appropriate? |
( ) |
( ) |
(x) |
( ) |
Are the methods adequately described? |
( ) |
( ) |
(x) |
( ) |
Are the results clearly presented? |
( ) |
( ) |
(x) |
( ) |
Are the conclusions supported by the results? |
( ) |
( ) |
(x) |
( ) |
Comments and Suggestions for Authors
Zhang et al. conducted a cross-sectional study involving 59,643 Chinese children aged 6-17 years. The aim of the study was to evaluate the effect of age and BMI groups (underweight, normal, overweight, obesity) on various markers of total body water and hydration status in each sex group. More specifically, they assessed total body water (TBW), intracellular water (ICW) and extracellular water (ECW). The authors concluded that overweight and obese children had significantly lower TBW/BW and higher TBW/FFM (FFM=Fat Free Mass) than those of normal weight. Therefore, they recommend that to maintain the stability of body water content, it is necessary to educate children on fluid intake according to their different characteristics, especially in the case of overweight children.
Major comments.
The data analysis is erratic. The authors indicate that differences in groups of variables were compared using analysis of variance (line 7), which is not developed in the statistical analysis subsection, and worse, is not really done. In addition, the way in which the analyses are performed (summary of the variables in means and standard deviations) leads to the conclusion that all variables in all groups are normally distributed, which is not justified. These issues are detailed in the following points.
- Table 2 is erratic for the following reasons:
- First: the authors have summarized all markers of total body water and hydration status in means and standard deviations. We understand that the authors have adequately tested the normality assumptions. Do all markers really follow a normal distribution in all age groups? In such a case, the simultaneous comparison between the four groups should be done with a single hypothesis test which, given normality, should be the F-test corresponding to the one-way ANOVA.
Response: Thanks for your comments.
We have checked all the data, and revised the presentation of the data that were with skewness distribution from means and standard deviations to the median and inter quartile ranges (IQR = 25th-75th percentile).
- If the F-test showed no significant differences, then pairwise comparisons are not appropriate. Otherwise, pairwise comparisons (six comparisons) should be made using a multiple comparisons test, for example, the Scheffee test. The use of the t-test is spurious (propagation of alphaerror).
Response: Thanks for your comments. We have revised accordingly in the Methods Section (Lines 141-153, Page 3).
- Table 3 is absolutely spurious. The authors intend to analyze the effect of age and BW (grouped in the three categories) on the markers of total body water and hydration status in means and standard deviations in each sex group. Given the normality of these markers, they should use an analysis of variance model (two-way ANOVA) with two factors of variance and interactions, which the authors have not done. Through this analysis, the authors should have tested for the following effects, namely: BW effect, age effect and the interaction between the two factors. Subsequently and when appropriate, the corresponding multiple comparisons should have been performed.
Response: Thanks for your comments.
We have made the correction accordingly (Lines 141-153, Page 3). Differences of variables in normal distribution including the proportions TBW/BW, ICW/TBW, ECW/TBW, ECW/ICW and TBW/FFM among the four age groups and three nutrition status groups were used one-way ANOVA; the Bonferroni correction for multiple tests was employed and the alpha level was set at 0.008 (0.05/6 comparisons) and 0.016 (0.05/3 comparisons), both in boys and girls. Differences of variables in skewness distribution such as the TBW, ICW, ECW and FFM among the four age groups and three nutrition status groups were assessed by Kruskal-Wallis H test, both in boys and girls. Mann-Whitney U test and Student's t test were used to compare the differences between sexes with the same age. Significance levels were set at 0.05 (p<0.05).
In our study, firstly, we divided the boys and girls into four age groups, respectively. We wanted to explore the mere effect of age on the total body water and its compartments, both in boys and girls. Furthermore, we also compared the values of total body water between different exes in the same age group, which was showed in Table 2.
Moreover, we also wanted to explore the mere effect of nutrition status, such as the BMI, on total body water and its compartments. Therefore, we divided the boys and girls into three BMI groups including the underweight, normal weight and overweight and obese, according to the evaluation criteria of different genders, respectively. Then, under each BMI group, we showed the values of boys and girls in four age groups, which was showed in Table 3.
Therefore, the results of the study showed that the age, sex and BMI influenced the total body water.
Minor comments
- Improve the following expression: “Data was presented as mean ± standard deviation and median and quartile ranges, if the data was normal distributed or not, respectively”. This one would be better: “Data was presented as mean and standard deviation (SD) when data followed a normal distribution, or as median and interquartile range (IQR = 25th – 75th percentile) when distribution departed from normality”.
Response: Thanks for your comments. We have made the correction accordingly (Lines 142-144, Page 3).
- Table 1. Change: “The characteristics of participants” to “Participant characteristics”. Likewise, use the thousands separator. For example, change “59643” to “59,643”.
Response: Thanks for your comments. We have made the correction accordingly (Line 159, Page 4).
- Figure 2 is inconsistent with the study design. For example, the authors have grouped the study subjects into four age groups, but in these figures age has been considered a continuous variable. In reality, a better analysis would have been to always use age on a continuous scale, if possible using at least one decimal place. This would have provided a broad age design that would have allowed the use of additive cubic spline models [1]. Such models are implemented in the mccv procedure corresponding to the R package [2]. The present figures are only a poor description of the data, as they do not provide whether or not the effects are significant.
Response: Thanks for your comments.
In our study, we divided the boys and girls into four age groups, respectively, according to the growth and development curves of themselves. Furthermore, relevant books were consulted in the classification of the age groups, such as the Dietary Reference Intake for China (DRIs) and the Dietary Guidelines for Chinese School-age Children. Therefore, we divided the boys and girls into pre-adolescence, adolescence, and post adolescence.
While, in order to present the forms of data in a better way, we showed the values in each age, in Figure 2, and we also wanted to provide more details for the researchers.
References as followed:
[1] Chinese Nutrition Society. Chinese dietary nutrients reference intakes (2023). Beijing: People's Health Publishing House; 2023.
[2] Chinese Nutrition Society. Chinese Dietary Guidelines for school-age Children (2022). Beijing: People's Health Publishing House; 2022.
- In expressions: Proportions of TBW/BW”, “ICW/TBW”, “ECW/TBW” and “TBW/FFM” the “of” is incorrect. Change to: Proportions “TBW/BW”, “ICW/TBW”, “ECW/TBW” and “TBW/FFM”.
Response: Thanks for your comments. We have made the correction accordingly (Line, Page).
- What is the style of the references used by the authors? It is certainly not Vancouver's.
Response: Thanks for your comments. It has been revised according to the style of the journal.
This study seems to be a continuation of another study in which the first author participates [3]. The present one seems promising because of the large sample size, but the statistical analysis should be redone by an expert.
References
[1] Hastie T, Tibshirani R. Generalized additive models for medical research. Stat Methods Med Res. 1995 Sep;4(3):187-96.
[2] R Core Team (2022). R: A language and environment for statistical computing. R Foundation for Statistical Computing, Vienna, Austria. URL <https://www.R-project.org/>.
[3] Zhang N, Zhang J, Du S, He H, Yan X, Ma G. Association between the content of intracellular and extracellular fluid and the amount of water intake among Chinese college students. Nutr Metab (Lond). 2019 Sep 18;16:67. doi: 10.1186/s12986-019-0397-9. PMID: 31548843; PMCID: PMC6751809.
Reviewer 2 Report
Comments and Suggestions for Authors
Jianfen Zhang investigated and reported the hydration status in more than 50 thousand children and found that the hydration status in overweight and obese children in China was lower.
The manuscript is well written, however, there are some issues that need to be addressed.
Abstract: Please define the abbreviations otherwise the abstract will not be stand-alone.
Introduction: the timeless facts should not be written in the past tense. Please adjust the language. Use past tense where you summarize the previous work.
Line-59-69, replace “Only several” with “Only a few”. Also add references of the studies which did report.
Methods: This section is very well written
Line 112- Correct “Obesity” to “Obese”.
Line- 115-118- this section should be moved to the end of the manuscript.
Results: Add significance * or p values in figures to indicate significant differences. Also, label each subfigure as A, B, C, etc. Then cite the figure in the results section for easy viewing.
Discussion: Needs a lot of revision. See below.
Line- 170-170- Is it consistent or inconsistent?
Line-171 and 176- Correct “Spain” spelling.
Line-180- replace “racial” with race”.
Line-180-181- should be a continuous sentence.
Line-213 and 217- Reference # 40 and 47, studies done in US not India or Indians.
Line-214- replace “different” with difference
Line-227- replace ‘Obesity” with “Obese”
Line-230-231- Which individuals? Please specify.
Line- 244- replace ‘nutritional status’ with “body composition”.
Line-247-248- Clarify the sentence “The mechanism that…… in obesity”.
Line 248-250- which study? Add the reference.
Line-254- change “access” to “assess”
Line- 254- change “influential factors” to “influence of age and BMI”
Line- 261- what do you mean by “rates of change of children”? Did you mean with age? Please clarify.
Line-263- insert “with” between those and normal.
Comments on the Quality of English LanguageThere are a lot of grammatical mistakes and sentences are unclear.
Author Response
Reviewer 2
Open Review
(x) I would not like to sign my review report
( ) I would like to sign my review report
Quality of English Language
( ) The quality of English does not limit my understanding of the research.
(x) The English could be improved to more clearly express the research.
|
|
Yes |
Can be improved |
Must be improved |
Not applicable |
Does the introduction provide sufficient background and include all relevant references? |
(x) |
( ) |
( ) |
( ) |
Is the research design appropriate? |
(x) |
( ) |
( ) |
( ) |
Are the methods adequately described? |
(x) |
( ) |
( ) |
( ) |
Are the results clearly presented? |
( ) |
(x) |
( ) |
( ) |
Are the conclusions supported by the results? |
( ) |
( ) |
(x) |
( ) |
Comments and Suggestions for Authors
Jianfen Zhang investigated and reported the hydration status in more than 50 thousand children and found that the hydration status in overweight and obese children in China was lower.
The manuscript is well written, however, there are some issues that need to be addressed.
Abstract: Please define the abbreviations otherwise the abstract will not be stand-alone.
Introduction: the timeless facts should not be written in the past tense. Please adjust the language. Use past tense where you summarize the previous work.
Line-59-69, replace “Only several” with “Only a few”. Also add references of the studies which did report.
Response: Thanks for your comments. It has been revised accordingly (Line 84, Page 2).
Methods: This section is very well written
Line 112- Correct “Obesity” to “Obese”.
Response: Thanks for your comments.
We checked the name of the Standard, and its name was the “Screening for overweight and obesity among school-age children and adolescents: WS/T 586-2018”.
Line- 115-118- this section should be moved to the end of the manuscript.
Response: Thanks for your comments. It has been moved to the end of the manuscript (Lines 350-352, Page 13).
Results: Add significance * or p values in figures to indicate significant differences. Also, label each subfigure as A, B, C, etc. Then cite the figure in the results section for easy viewing.
Response: Thanks for your comments.
In our study, we divided the boys and girls into four age groups, respectively, according to the growth and development curves of themselves. Furthermore, relevant books were consulted in the classification of the age groups, such as the Dietary Reference Intake for China (DRIs) and the Dietary Guidelines for Chinese School-age Children. Therefore, we divided the boys and girls into pre-adolescence, adolescence, and post adolescence, and the values were showed in Table 2.
Moreover, we divided the boys and girls into three BMI groups, and the values were showed in Table 3.
In order to better show the changes of values in each age and each BMI, we demonstrated the values in Figure 2 and Figure 3, but we did not comparing the differences, just to show the changes of the values in a more intuitive method.
Discussion: Needs a lot of revision. See below.
Line- 170-170- Is it consistent or inconsistent?
Response: Thanks for your comments. It was “in consistent with” (Line 237, Page 11).
Line-171 and 176- Correct “Spain” spelling.
Response: Thanks for your comments. It has been revised accordingly (Line 238, Page 11).
Line-180- replace “racial” with race”.
Response: Thanks for your comments. It has been revised accordingly (Line 247, Page 11).
Line-180-181- should be a continuous sentence.
Response: Thanks for your comments. It has been revised accordingly (Lines 246-250, Page 11).
Line-213 and 217- Reference # 40 and 47, studies done in US not India or Indians.
Response: Thanks for your comments. It has been revised accordingly (Lines280, 283 and 289, Page11-12).
Line-214- replace “different” with difference
Response: Thanks for your comments. It has been revised accordingly (Line 285, Page 11).
Line-227- replace ‘Obesity” with “Obese”
Response: Thanks for your comments. It has been revised accordingly (Line 299, Page 12).
Line-230-231- Which individuals? Please specify.
Response: Thanks for your comments. It has been revised accordingly, by adding “children, adolescents and healthy adults” (Lines 303-304, Page 12).
Line- 244- replace ‘nutritional status’ with “body composition”.
Response: Thanks for your comments.
According to some references, the nutrition status including the BMI, and in our study, we only analyzed the effect of BMI on the total body water.
Line-247-248- Clarify the sentence “The mechanism that…… in obesity”.
Response: Thanks for your comments. It has been revised accordingly “This might be due to the fact that the excessive body weight (BW) stemmed from a relatively high proportion of body fat as compared to the proportions of FFM and TBW in relation to BW, even though the TBW and FFM increased with BMI in all age groups” (Lines 295-298, Page 12).
Line 248-250- which study? Add the reference.
Response: Thanks for your comments. It has been revised accordingly, and we added the reference of 60 into the manuscript (Line 325, Page 12).
Line-254- change “access” to “assess”
Response: Thanks for your comments. It has been revised accordingly (Line 328, Page 12).
Line- 254- change “influential factors” to “influence of age and BMI”
Response: Thanks for your comments. It has been revised accordingly, by revising the “influential factors” to “assess the influential factors including the age, sex and nutrition status” (Lines 328-329, Page 12).
Line- 261- what do you mean by “rates of change of children”? Did you mean with age? Please clarify.
Response: Thanks for your comments. It has been revised accordingly, by revising the “rates of changes” into “changes” (Line 338, Page 12).
Line-263- insert “with” between those and normal.
Response: Thanks for your comments. It has been revised accordingly (Line 340, Page 12).
Comments on the Quality of English Language
There are a lot of grammatical mistakes and sentences are unclear.
Response: Thanks for your comments. It has been revised accordingly across the manuscript.
Round 2
Reviewer 1 Report
Comments and Suggestions for Authors
Major comments
In this revised version, the authors have corrected some of the errors noted in the previous version. However, the data analysis is flawed with respect to multiple comparisons. These flaws were pointed out in my previous review (point 2 of major comments).
The data analysis in this research basically consists of comparing the distributions of numerical variables between groups, namely: overall age groups, age groups according to body mass groups and sex groups. When the numerical variables in each study group satisfy the assumptions of normality and hoccedasticity, the optimal test is the F-test corresponding to the analysis of variance with a factor of variation (ANOVA-I). The p-value corresponding to this simultaneous comparison should be specified in the last column of the corresponding table. The authors should understand that contrasts with p-values of, for example, 0.08 and 0.80 are not significant, but the reader can appreciate that in the first case the data point to significance, which is not the case with p=0.80.
Assuming that the contrast based on the F-test is not significant (p > 0.05), it is not appropriate to carry out multiple comparisons, since the p-value of the F-test prevails over the multiple comparisons tests. When the contrast is significant, it is appropriate to carry out multiple comparisons. The authors indicate that they use the Bonferroni test for multiple comparisons, but this does not appear to be shown in the tables. When they say that there are differences between groups it is not clear whether they are referring to the F-test or the Bonferroni test, which would indicate between which groups there are significant differences. However, under the assumptions of normality and homoscedasticity there are better methods; for example, Scheffe's method of multiple comparisons. To express the results of multiple comparisons, the authors should use above each mean (SD) an alphabetical superscript indicating between which groups there is a significant difference according to the multiple comparisons method used. I suggest that the authors google the expression “different superscripts indicate differences ...”. They will be able to see in this way the correct way to express multiple comparisons.
This procedure would be identical when the above assumptions of normality and homoscedasticity are not satisfied but replacing the F-test (ANOVA-1) by the Kruskal-Wallis test.
To avoid the resulting table being easily readable, comparisons between sex groups should be expressed by the corresponding p-values in the last row of the table. Otherwise there would be confusion between multiple comparisons by age groups with comparisons between sex groups.
Minor comments
Lines 132-134. The definition of the variable BMI should not be included in the statistical methods subsection, but in a specific variable definition section.
Line 148. Change: “compare differences between sexes” to “compare differences between groups of sexes”.
I think the authors should consult with an expert statistician for data analysis.
Author Response
Open Review
( ) I would not like to sign my review report
(x) I would like to sign my review report
Quality of English Language
(x) The quality of English does not limit my understanding of the research.
( ) The English could be improved to more clearly express the research.
|
|
Yes |
Can be improved |
Must be improved |
Not applicable |
Does the introduction provide sufficient background and include all relevant references? |
(x) |
( ) |
( ) |
( ) |
Is the research design appropriate? |
( ) |
(x) |
( ) |
( ) |
Are the methods adequately described? |
( ) |
( ) |
(x) |
( ) |
Are the results clearly presented? |
( ) |
( ) |
(x) |
( ) |
Are the conclusions supported by the results? |
( ) |
( ) |
(x) |
( ) |
Comments and Suggestions for Authors
Major comments
In this revised version, the authors have corrected some of the errors noted in the previous version. However, the data analysis is flawed with respect to multiple comparisons. These flaws were pointed out in my previous review (point 2 of major comments).
The data analysis in this research basically consists of comparing the distributions of numerical variables between groups, namely: overall age groups, age groups according to body mass groups and sex groups. When the numerical variables in each study group satisfy the assumptions of normality and hoccedasticity, the optimal test is the F-test corresponding to the analysis of variance with a factor of variation (ANOVA-I). The p-value corresponding to this simultaneous comparison should be specified in the last column of the corresponding table. The authors should understand that contrasts with p-values of, for example, 0.08 and 0.80 are not significant, but the reader can appreciate that in the first case the data point to significance, which is not the case with p=0.80.
Assuming that the contrast based on the F-test is not significant (p > 0.05), it is not appropriate to carry out multiple comparisons, since the p-value of the F-test prevails over the multiple comparisons tests. When the contrast is significant, it is appropriate to carry out multiple comparisons. The authors indicate that they use the Bonferroni test for multiple comparisons, but this does not appear to be shown in the tables. When they say that there are differences between groups it is not clear whether they are referring to the F-test or the Bonferroni test, which would indicate between which groups there are significant differences. However, under the assumptions of normality and homoscedasticity there are better methods; for example, Scheffe's method of multiple comparisons. To express the results of multiple comparisons, the authors should use above each mean (SD) an alphabetical superscript indicating between which groups there is a significant difference according to the multiple comparisons method used. I suggest that the authors google the expression “different superscripts indicate differences ...”. They will be able to see in this way the correct way to express multiple comparisons.
This procedure would be identical when the above assumptions of normality and homoscedasticity are not satisfied but replacing the F-test (ANOVA-1) by the Kruskal-Wallis test.
To avoid the resulting table being easily readable, comparisons between sex groups should be expressed by the corresponding p-values in the last row of the table. Otherwise there would be confusion between multiple comparisons by age groups with comparisons between sex groups.
Response: Thanks for your comments. It has been revised accordingly in the manuscript, including the Methods Section and the Results Section. We have used the Scheffe's method of multiple comparisons to comparing the differences among different ages groups and BMI groups. Furthermore, we added p-values in the right column of the table, and when the one-way ANOVA and Kruskal-Wallis H test were significant, we compared the differences between different age groups or BMI groups. Moreover, different alphabetical superscripts were used to show the differences between different groups, such as “a: significant differences were found when comparing with age 6-8 years group; b: significant differences were found when comparing with age 9-11 years group; c: significant differences were found when comparing with age 12-14 years group; d: significant differences were found when comparing with age 15-17 years group”.
Minor comments
Lines 132-134. The definition of the variable BMI should not be included in the statistical methods subsection, but in a specific variable definition section.
Response: Thanks for your comments. We have revised accordingly in the Methods Section (Lines 118-119, Page 3).
Line 148. Change: “compare differences between sexes” to “compare differences between groups of sexes”.
Response: Thanks for your comments. We have made the correction accordingly (Line 153, Page 3).
I think the authors should consult with an expert statistician for data analysis.
Reviewer 2 Report
Comments and Suggestions for Authors
Manuscript has been appropriately revised and comments are addressed.
Author Response
Point-by-point response to comments
Reviewer 2
Open Review
(x) I would not like to sign my review report
( ) I would like to sign my review report
Quality of English Language
(x) The quality of English does not limit my understanding of the research.
( ) The English could be improved to more clearly express the research.
Yes
Can be improved
Must be improved
Not applicable
Does the introduction provide sufficient background and include all relevant references?
(x) ( ) ( ) ( )
Is the research design appropriate?
(x) ( ) ( ) ( )
Are the methods adequately described?
(x) ( ) ( ) ( )
Are the results clearly presented?
(x) ( ) ( ) ( )
Are the conclusions supported by the results?
(x) ( ) ( ) ( )
Comments and Suggestions for Authors
Manuscript has been appropriately revised and comments are addressed.
Response: Thanks for your comments.
Round 3
Reviewer 1 Report
Comments and Suggestions for Authors
The authors have enhanced the statistical analysis in this version compared to the previous one. Now the authors make a simultaneous comparison of means with the F-test or of medians with the Kruskal-Wallis test and when the test is significant they use the multiple comparisons methods. In the case of means they use the Scheffe test, which is preferable to the Bonferroni test (the latter is very conservative). In addition, there are grammatical errors that need to be corrected. Table formats should also be improved.
Coments
1. Throughout the text, separate syllables appear in line breaks. In English, syllables are separated according to pronunciation. Thus, separations such as the one at the end of line 66 are incorrect (numer- ous) or at the end of line 111 (partic- ipants). The authors should make the corresponding corrections.
2. Line 115. The expression “Body mass index (BMI) was assessed with weight and height as kg/m2 ” is poor. It would look better as follows: the body mass index (BMI) was defined as the ratio of weight (kg) to the square of height (m2).
3. Section 2.3. What method of multiple comparisons have you used for non-normal variables? Obviously it could not be Scheffe's, which is based on normality and homocedasticity. The method of Games-Howell for multiple comparison is implemented in SAS and R packages.
4. Line 133. Change: “age9-11” by “age 9-11”.
5. Line 140. Change “inter quartile” by “interquartile”.
6. The authors should improve the formatting of tables two and three. For example, the totals column in both tables are unacceptable.
7. Lines 143-144. The authors seem not to have understood Scheffe's method for multiple comparisons. When applied, the result is a triangular matrix of p-values. These are the p-values that are reported. They should not be penalized. Here Scheffe's method is mixed with Bonferroni's method.
8. Line 150: Change “Significance levels were set at 0.05 (p<0.05)” to “Statistical significance was set at p < 0.05”
9. Table 2. The way in which multiple comparisons are expressed by superscripts is atypical and awkward to interpret. The common and straightforward method is that two groups with no significant differences share the same superscript. Thus, the sequence (table 2): “bdc, acd, abd, abc” changes to: “a, b, c, d”. I recommend the authors to google scholar for the term: “Different superscripts indicate significant differences”.
10. Line 174. Authors say: “Values are shown as the mean±standard deviation (SD)”. This expression is not complete, since in the table there are variables that are summarized in means and SDs and others in medians and IQRs. It should therefore read: “Values are shown as the mean±standard deviation (SD) and medians (IQR)”.
11. Line 196. Delete the term “while”. It should not be at the beginning of a sentence.
12. Lines 240-241. The phrase: “The content of TBW, ICW and ECW among children aged in Portugal (26.7 kg, 15.1 kg and 11.6 kg)”,… is erratic. Please check it out.
13. As indicated in my previous review, I think the authors should consult with an expert statistician for data analysis.
Author Response
Point-by-point response to comments
Reviewer 1
Open Review
( ) I would not like to sign my review report
(x) I would like to sign my review report
Quality of English Language
(x) The quality of English does not limit my understanding of the research.
( ) The English could be improved to more clearly express the research.
Yes Can be improved Must be improved Not applicable
Does the introduction provide sufficient background and include all relevant references?
(x) ( ) ( ) ( )
Is the research design appropriate?
(x) ( ) ( ) ( )
Are the methods adequately described?
( ) ( ) (x) ( )
Are the results clearly presented?
( ) ( ) (x) ( )
Are the conclusions supported by the results?
(x) ( ) ( ) ( )
Comments and Suggestions for Authors
The authors have enhanced the statistical analysis in this version compared to the previous one. Now the authors make a simultaneous comparison of means with the F-test or of medians with the Kruskal-Wallis test and when the test is significant they use the multiple comparisons methods. In the case of means they use the Scheffe test, which is preferable to the Bonferroni test (the latter is very conservative). In addition, there are grammatical errors that need to be corrected. Table formats should also be improved.
Comments
- Throughout the text, separate syllables appear in line breaks. In English, syllables are separated according to pronunciation. Thus, separations such as the one at the end of line 66 are incorrect (numer- ous) or at the end of line 111 (partic- ipants). The authors should make the corresponding corrections.
Response: Thanks for your comments. It has been revised accordingly across the manuscript.
- Line 115. The expression “Body mass index (BMI) was assessed with weight and height as kg/m2 ” is poor. It would look better as follows: the body mass index (BMI) was defined as the ratio of weight (kg) to the square of height (m2).
Response: Thanks for your comments. We have revised accordingly in the Methods Section (Lines 118-119, Page 3).
- Section 2.3. What method of multiple comparisons have you used for non-normal variables? Obviously it could not be Scheffe's, which is based on normality and homocedasticity. The method of Games-Howell for multiple comparison is implemented in SAS and R packages.
Response: Thanks for your comments. Differences of variables in skewness distribution such as the TBW, ICW, ECW and FFM among the four age groups and three nutrition status groups were assessed by Kruskal-Wallis H test, both in boys and girls; and the alpha level was set at 0.008 (0.05/6 comparisons) and 0.017 (0.05/3 comparisons) for multiple tests.
- Line 133. Change: “age9-11” by “age 9-11”.
Response: Thanks for your comments. It had been revised accordingly (Line 136, Page 3).
- Line 140. Change “inter quartile” by “interquartile”.
Response: Thanks for your comments. We have made the correction accordingly (Line 143, Page 3).
- The authors should improve the formatting of tables two and three. For example, the totals column in both tables are unacceptable.
Response: Thanks for your comments. We have revised accordingly in table 2 and table 3.
- Lines 143-144. The authors seem not to have understood Scheffe's method for multiple comparisons. When applied, the result is a triangular matrix of p-values. These are the p-values that are reported. They should not be penalized. Here Scheffe's method is mixed with Bonferroni's method.
Response: Thanks for your comments.
We have discussed about the statistical method you recommended and we added the details in the Methods Section (Lines 141-153, Page 3) and Results Section (Lines 281-289, Page 5). The Scheffe's method for multiple tests was employed and the alpha level was set at 0.05.
- Line 150: Change “Significance levels were set at 0.05 (p<0.05)” to “Statistical significance was set at p < 0.05”
Response: Thanks for your comments. It had been revised accordingly (Lines 153-154, Page 3).
- Table 2. The way in which multiple comparisons are expressed by superscripts is atypical and awkward to interpret. The common and straightforward method is that two groups with no significant differences share the same superscript. Thus, the sequence (table 2): “bdc, acd, abd, abc” changes to: “a, b, c, d”. I recommend the authors to google scholar for the term: “Different superscripts indicate significant differences”.
Response: Thanks for your comments. We have made the correction accordingly under the table that “a: There were significant differences between age 6-8 years group and age 9-11 years group (p<0.05 for normal data and p<0.008 for data in skewness distribution); b: there were significant differences between age 6-8 years group and age 12-14 years group (p<0.05 for normal data and p<0.008 for data in skewness distribution); c: there were significant differences between age 6-8 years group and age 15-17 years group (p<0.05 for normal data and p<0.008 for data in skewness distribution); d: there were significant differences between age 9-11 years group and age 12-14 years group (p<0.05 for normal data and p<0.008 for data in skewness distribution); e: there were significant differences between age 9-11 years group and age 15-17 years group (p<0.05 for normal data and p<0.008 for data in skewness distribution); f: there were significant differences between age 12-14 years group and age 15-17 years group (p<0.05 for normal data and p<0.008 for data in skewness distribution)”.
- Line 174. Authors say: “Values are shown as the mean±standard deviation (SD)”. This expression is not complete, since in the table there are variables that are summarized in means and SDs and others in medians and IQRs. It should therefore read: “Values are shown as the mean±standard deviation (SD) and medians (IQR)”.
Response: Thanks for your comments. We have made the correction accordingly (Line 143, Page 3).
- Line 196. Delete the term “while”. It should not be at the beginning of a sentence.
Response: Thanks for your comments. It had been revised accordingly (Line 211, Page 6).
- Lines 240-241. The phrase: “The content of TBW, ICW and ECW among children aged in Portugal (26.7 kg, 15.1 kg and 11.6 kg)”,… is erratic. Please check it out.
Response: Thanks for your comments. We have revised accordingly (Line 261, Page 10).
- As indicated in my previous review, I think the authors should consult with an expert statistician for data analysis.
Response: Thanks for your comments. We have consulted the expert statistician for data analysis and revised the related details in the manuscript.